# Using Femtosecond Laser Pulses to Explore the Nonlinear Optical Properties of Au NP Colloids That Were Synthesized by Laser Ablation

**DOI:** 10.3390/nano12172980

**Published:** 2022-08-28

**Authors:** Mohamed Ashour, Hameed G. Faris, Hanan Ahmed, Samar Mamdouh, Kavintheran Thambiratnam, Tarek Mohamed

**Affiliations:** 1Laser Institute for Research and Applications LIRA, Beni-Suef University, Beni-Suef 62511, Egypt; 2High Institute of Optics Technology HIOT, Sheraton Heliopolis, Cairo 11799, Egypt; 3Al Anbar Health Directorate, Training and Human Development Center, Ramadi 31001, Iraq; 4Photonics Research Centre, University of Malaya, Kuala Lumpur 50603, Malaysia; 5Department of Engineering, Faculty of Advanced Technology and Multidiscipline, Universitas Airlangga, Surabaya 60115, Indonesia

**Keywords:** nonlinear optics, nanoparticles, femtosecond laser, high repetition rate, gold nanoparticles, nonlinear absorption, nonlinear refraction

## Abstract

In this study, we experimentally investigated the nonlinear optical properties of Au nanoparticles (Au NPs) that were prepared in pure distilled water using the laser ablation method. The Au NPs were prepared using a nanosecond Nd:YAG laser with an ablation time of 5 or 10 min at a constant laser energy of 100 mJ. The structure and the linear optical properties of the Au NPs were investigated using a transmission electron microscope (TEM) and UV-visible spectrophotometer analysis, respectively. The TEM measurements showed that the average size of the Au NPs varied from 20.3 to 14.1 nm, depending on the laser ablation time. The z-scan technique was used to investigate the nonlinear refractive index ($n_{2}$) and nonlinear absorption coefficient (γ) of the Au NPs, which were irradiated at different excitation wavelengths that ranged from 740 to 820 nm and at different average powers that ranged from 0.8 to 1.6 W. The Au NP samples exhibited a reverse saturable absorption (RSA) behavior that increased when the excitation wavelength and/or incident laser power increased. In addition, the Au NPs acted as a self-defocusing material whenever the excitation wavelength or incident power were modified.

## 1. Introduction

The plethora of recently developed and potential future applications for new nonlinear optical (NLO) materials in optics and photonics has attracted the interest of many researchers, particularly nanosized NLOs. This is because nanostructured NLO materials inherently have various unique and highly desirable characteristics, such as high surface/volume ratios, unique structures, and better optical and electrical properties than their bulkier counterparts [1,2,3].

Among the different nanostructure materials, metal nanoparticles (MNPs) have been found to have substantial uses in a very wide range of applications, such as biosensors [4], cancer therapy [5], waveguiding [6], data storage [7], all-optical switching devices [8], and many more. Moreover, noble metallic nanoparticles (NMNPs) offer easy chemical synthesis, high stability, and tunable surface functionalization [9]. In particular, silver (Ag) and gold (Au) NPs are among the most useful nanomaterials out of the NMNPs, especially for optoelectronic and nonlinear optical applications [10].

Gold nanoparticles (Au NPs) have attracted a great deal of interest in many fields over the past few years due to their strong surface plasmon resonance (SPR) in the visible spectral region. The properties of Au NPs can be tuned by controlling their shape, size, and degree of aggregation and the local environment [11]. Au NPs have unique applications as catalyses [12], diagnosis biomarkers [13], biological imaging [14], colorimetric sensors [15], and electronics [16].

The rapid development of different techniques for producing Au NPs with different shapes and sizes has provided several new opportunities for nonlinear optics. Laser ablation is one of the most promising methods that can be used to produce NPs as it a cost-effective, dependable, and economical method that allows the remarkable nonlinear optical properties of NPs to be demonstrated [17].

Many studies on the linear and nonlinear optical properties of Au NPs have used different methods to prepare NPs of different sizes and shapes [18,19,20,21,22,23,24]. It can be clearly seen from these reports that the NLO properties of Au NPs are strongly related to the size and the shape of the nanoparticles, as well as the properties of the selected laser beam.

Although there have been a few studies on the NLO properties of Au NPs using the z-scan approach [18,19,20,21,22,23,24], no systematic studies on the NLO properties of Au NP colloids using femtosecond laser pulses for different sizes and concentrations of Au NPs and different laser parameters, such as excitation wavelength and intensity, have been conducted to our knowledge. For instance, the authors of [18,19,20,22,23] studied the NLO properties of Au NPs at a certain excitation wavelength and for a given Au NP size. In [24], the NLO properties of Au NPs were studied at various concentrations and at a certain excitation wavelength.

The present study was carried out to investigate the NLO properties of Au NPs that were prepared using the laser ablation method with a nanosecond Nd:YAG laser source. The nonlinear absorption coefficient (γ) and nonlinear refractive index ($n_{2}$) of the NPs were investigated experimentally using both the open- (OA) and closed-aperture (CA) z-scan techniques with femtosecond laser pulses. The Au samples were prepared at two different laser ablation times (LAT): 5 min and 10 min. The NLO properties of the Au NPs were measured by exposing the samples once at different excitation wavelengths that ranged from 740 to 820 nm at a constant incident power of 1.4 W and once at different incident average powers that ranged from 0.8 to 1.6 W at a constant wavelength of 800 nm.

## 2. Experimental Setup

### 2.1. Laser Ablation Setup

Among the many different techniques for the synthesis of Au NPs [25,26,27], the laser ablation technique is believed to be the most efficient, reliable, and cost-effective, as well as the best in the terms of the size distribution of the nanoparticles. Figure 1 shows the experimental setup that was used to produce the Au NPs in this study. A Quanta-Ray PRO-Series 350-10 nanosecond 532 nm Nd:YAG laser source with a 10 Hz repetition rate and a pulse duration of 10 ns from Spectra-Physics was used for the synthesis of the Au NPs.

The Au NP colloids were created by exposing a square piece of bulk Au with a purity of about 99% to the laser beam while the sample was immersed in a glass beaker that was filled with 22 mL of distilled water. The laser beam was guided by three mirrors and then focused onto the bulk Au sample using a 10 cm convex lens, for which a glass cap with a hole with a diameter that was close to the diameter of the laser beam was used to avoid the occurrence of water splashing as this could affect the quality of the laser ablation output. The LATs of 5 and 10 min were enough to obtain Au NP colloids without agglomeration and with good solubility using magnetic stirrers.

### 2.2. Z-Scan Setup

There are many techniques that can be used to measure the NLO properties of materials, such as degenerate four-wave mixing, optical Kerr gate, z-scans, and nonlinear interferometry [28,29]. Overall, the z-scan technique [30,31] is considered to be a highly sensitive single-beam technique, which uses the principle of spatial beam distortion to measure both the sign and magnitude of the nonlinear optical coefficients; thus, it was the most suitable technique for this study. Figure 2 shows the z-scan setup that was used in this work, which implemented both the OA and CA methods. During the experiment, the Au NPs were irradiated using an INSPIRES HF100 intense tunable pulsed femtosecond laser The spatial profile of this beam had a Gaussian distribution with a TEM_00_ spatial mode and M^2^ < 1.1. This system covered a wavelength range of 345–2500 nm with an 80 MHz repetition rate and a 100 fs pulse duration. This laser was pumped by a MAI-TAI HP mode-locked femtosecond Ti:Sapphire laser that operated in a wavelength range between 690 nm and 1040 nm with an 80 MHz repetition rate. Both laser systems were procured from Spectra-Physics Inc., Milpitas, CA, USA.

In this experiment, the beam was passed through an attenuator that controlled the incident power of the laser and then through a convex lens with a 5 cm focal length to focus the laser beam on the measured sample. The Au NP samples were placed in micro quartz cuvettes that had a volume of 350 µL and dimensions of 52 mm × 1.5 mm × 3.5 mm, in which the samples were translated ± 2.5 cm around the lens focus in z directions. The cuvettes had a 1 mm path length, which ensured that the Au NP samples were thin. A 50/50 beam splitter was used to measure the nonlinear refractive index $n_{2}$ and the nonlinear absorption coefficient γ simultaneously.

In the OA method, the whole transmission was measured a power meter (PM2), which meant that small distortions became insignificant and signal variations were just due to the nonlinear absorption. In the CA method, the beam that focused on the samples passed through a closed aperture that was placed in front of a power meter (PM1) to ensure that if the beam experienced any nonlinear phase shifts, they were only due to variations in the samples. In both setups, the same model of power meter was used (Newport 843 R power meter).

The linear normalized transmittance value (S) should be 0.1<S<0.5, so this value was set as S=1 and S=0.3 in the OA and CA measurements, respectively. Note that the sample thickness (L) met the required condition of $L\ll Z_{0}/\Delta \Phi$, where $Z_{0}$ is the Rayleigh range [31].

## 3. Results and Discussion

### 3.1. Structure and Size Distribution of the Au NPs

A transmission electron microscope (TEM) was used to investigate the structures of the Au NPs that were obtained using the laser ablation method and determine their average size. Figure 3 shows the size distribution of the Au NPs that were prepared at an LAT of 5 min and a laser energy of 100 mJ, for which the average NP size was determined to be about 20.3 nm. Figure 4 shows the distribution of the Au NPs that were prepared at an LAT of 10 min and a laser energy at 100 mJ, for which the average size of the Au NPs reduced from 20.3 to 14.1 nm.

The concentration of Au NPs was measured for both samples using an Agilent Technologies 200 series AA atomic absorption spectrometer(Agilent, Santa Clara, CA, USA) and were determined to be 4 mg/L and 8 mg/L for the 5 min and 10 min LAT samples, respectively. The TEM images of the Au NPs in the colloidal solutions that were prepared in distilled water are shown as insets in Figure 3 and Figure 4. The images show that the Au NPs were spherical. As a result of the fragmentation mechanism, it could be concluded that the greater the LAT, the smaller the nanoparticles [32].

Based on the obtained topography for the Au NP samples, the volume-specific surface area (VSSA) method [33,34] was used to calculate the total surface area of all of the NPs in 22 mL of distilled water using Equation (1). Knowing that the density of Au is $\rho =19.32\;g/{\mathrm{cm}}^{3},\;$ the total surface area was found to be $33.67\;m^{2}/g$ and $48.454\;m^{2}/g$ for the 5 min and 10 min LAT samples, respectively.
(1)SA=N×sa 
where *SA* is the total surface area of NPs per volume, *sa* is the surface area per particle ($sa=4\pi r^{2}$, where r is the average radius), and $N=\frac{V}{v}$ is the total number of NPs (where V=m/ρ is the volume of the total NPs and $v\;=\left(4/3\right)\pi r^{3}$ is the volume of one particle).

### 3.2. Linear Optical Properties of the Au NPs

The linear optical transmission of the Au NPs was measured using a UV-VIS spectrophotometer with a wavelength range of 200–800 nm. The absorption spectrum in Figure 5 shows that two absorption peaks appeared in both samples: one in the visible region, which was due to the localized surface plasmon resonance (LSPR) absorption, and another in the UV region, which was due to the interband transition [35]. The LSPR peaks in the visible region are shown in the inset of Figure 5.

The obtained spectrum was used to calculate the energy bandgap of the Au NP samples using the Tauc plot method [36,37], in which the absorption spectrum that was measured using the spectrophotometer was extrapolated. The energy bandgap *E*_ɡ_ of both Au NP samples was estimated by fitting the Tauc equation for direct bandgap material to Equation (2):(2)$${\left(\alpha hv\right)}^{2}=A\left(hv-E_{g}\right)\;$$
where α is the linear absorption coefficient, hv is the photon energy, and A is an energy-independent constant. Figure 6 shows that increasing the LAT from 5 min to 10 min led to a decrease in the energy bandgap of the Au NPs from 5.42 to 5.34 eV. The blue line shows the shift in the SPR peak for the 10 min sample in comparison to that of the 5 min sample, which indicated that the average size of the NPs decreased [35,38] (which was also proved by Figure 3 and Figure 4). It is also worth mentioning that due to the high value of the VSSA for the 10 min LAT sample, the absorbance was higher compared to that of the 5 min LAT sample. In general, decreasing the nanoparticle size led to an increase in the energy bandgap because of the electron confinement theory at the nanoscopic scale. However, this was only true for NPs within the quantum range of (1–10 nm and the opposite occurred when the particle size was larger than 10 nm, as shown in Figure 6 [35].

### 3.3. Nonlinear Optical Properties of the Au NPs

#### 3.3.1. OA Measurements

As discussed in Section 2.2, the OA setup was used to measure the nonlinear absorption coefficient γ of the prepared Au NP samples. Both samples that were obtained using LATs of 5 and 10 min exhibited RSA behavior, i.e., they had positive nonlinear absorption coefficient. The normalized transmittance was expressed by Equation (3) [30,31]:(3)$$\Delta T_{OA}=1\pm \frac{\gamma I_{0}L_{eff}}{n^{\left(3/2\right)}\left[1+{\left(z/z_{o}\right)}^{2}\right]}\;$$
where *z* is the relative position of the sample, *I*_0_ is the laser peak intensity, and $z_{o}$ is the Rayleigh length, which was calculated using $z_{o}=\pi w_{0}{}^{2}/n\lambda$ (where $w_{o}$ is the laser beam waist (16.8 µm ± 0.17) and $L_{eff}$ is the effective length of the sample, which was given by $L_{eff}=\left(1-e^{-n\alpha_{0}L}\right)/n\alpha_{0},$ where *n* = 1 for two-photon absorption (2PA) and *n* = 2 for three-photon absorption (3PA)). In this study, the best fit was achieved by 3PA, which was found to match the experimental results.

During this measurement, the dependency of the nonlinear absorption coefficient on both the excitation wavelength and the incident power was also investigated. The samples exhibited both excitation wavelength and power dependency. Figure 7a,b show the dependency of the nonlinear absorption coefficient on the excitation wavelength of the Au NP samples at concentrations of 4 and 8 mg/L, respectively. The excitation wavelength varied from 740 nm to 820 nm at a constant average incident power of 1.4 W.

Figure 8 shows the relationship between the excitation wavelength and the nonlinear absorption coefficient γ for the 5 and 10 min LAT samples at a constant average power of 1.4 W. Figure 9a,b shows the nonlinear absorption coefficient γ measurements for both Au NP samples with varying incident power values $P_{\mathrm{avg}}$ (from 0.8 to 1.6 W) at a constant excitation wavelength of 800 nm. The results of this measurement appeared to be analogous to those of the measurement in which the incident power was constant.

Figure 10 shows the behavior of the samples when the NL absorption coefficient γ linearly increased with the increasing incident power.

The results of the OA z-scan that are shown in Figure 7 and Figure 9 could be explained as follows. When exposed to a certain excitation wavelength or excitation power, the Au NPs exhibited a strong absorption, especially when the samples approached the focal point of the convex lens, whereupon the normalized transmittance increased gradually and the samples exhibited RSA. The qualitative analysis proved this numerically, as illustrated in Figure 8 and Figure 10. The results also showed that whatever the concentration of the Au NPs, the value of the NL absorption coefficient γ increased with the increasing excitation wavelength or excitation power, which meant that at higher excitation wavelengths or powers, the excited state of the samples had a strong absorption compared to the ground state. This behavior could be due to free-carrier absorption in which electron or hole transitions occurred by absorbing free photons [39,40] or the 3PA process, whereby three photons were absorbed simultaneously by the atom and the electron moved from a lower energy state to a higher energy state. This behavior could also be due to both effects [41].

Furthermore, the possibility of the photodegradation of the nanostructures, the light scattering process, and the hot electron nonlinearity could be reasons for the presence of RSA [41]. However, hot electron nonlinearity in nanoparticles requires a few picoseconds to arise [41]. In our measurements, the selected excitation source was a femtosecond laser; thus, hot electrons were not responsible for optical nonlinearity.

By knowing the value of the NL absorption coefficient γ, the 3PA cross-section $\sigma_{3}$ could be calculated using Equation (4) [42]:(4)$$\sigma_{3}=\frac{\gamma }{N_{A}d_{o}10^{-3}}{\left(\frac{hc}{\lambda }\right)}^{2}\;$$
where $N_{A}$ is the Avogadro’s number, $d_{o}\;\mathrm{mol}/L$ is the concentration of Au NPs, and hc/λ is the energy of the incident photons. The values of $\sigma_{3}$ for the two samples are shown in Table 1. It was obvious that the results of the 3PA cross-section matched with the results that were obtained for the NL absorption coefficients of both samples.

Table 1 shows that as the concentration of Au NPs decreases, the 3PA cross-section increased and that the 3PA cross-section increased when both the wavelength and average power increased.

#### 3.3.2. CA Measurements

Figure 11 and Figure 12 show the CA measurements in which the nonlinear refractive index $n_{2}$ was measured for the same Au NPs samples. The results showed that the samples acted as a self-defocusing material that had a negative refractive index. The CA measurements manifested that whenever the average power or excitation wavelength increased, the peak valley difference $T_{P-V}$ increased too. Increasing the nanoparticle concentration from 4 mg/L to 8 mg/L at the same laser parameter led to a decrease in the $T_{P-V}$ value.

When the fact that the used laser had a high repetition rate (HRR) of 80 MHz was taken into consideration, this could have led to the formation of a thermal lens in the samples due to the thermal effect [43]. This effect happens when the temperature increases in a material due to the conversion of absorbed light into heat energy, which results in a change in the material density and, subsequently, the nonlinear refractive index of the sample [43,44].

The results of the CA measurements that are shown in Figure 11 and Figure 12 proved the conformity between the theoretical transmittance curves, which are presented as solid lines and the experimental data are presented as dots. The CA results indicated a decrease in the nonlinear refractive index with respect to the nanoparticle size, which was due to the increasing number of particles in the medium, as well as the increasing ratio of linear absorption to thermal diffusivity within the medium [44]. Generally, the thermal effect in the samples increased when the time interval between the laser pulses was shorter than the thermal diffusion time of the sample, i.e., when $t_{c}=w/4d$, where w is the beam diameter and $d=k/\rho c_{p}$ is the thermal diffusion coefficient (where k is the thermal conductivity, $c_{p}$ is the specific heat capacity, and ρ is the density of the Au NPs). The selected femtosecond laser had a time of about 12.5 ns between the laser pulses, which was much shorter than the thermal diffusion time $t_{c}$ of the Au NPs [45,46]. This meant that the samples were not able to return to their equilibrium state within the time between the laser pulses.

Some reports have suggested different mathematical models that could overcome the use of HRR lasers by separating the Kerr nonlinearity from that of the thermo-optical effect [19,47,48]. In this study, the miasmatical model that was suggested by [48] was used and the thermal focal length was calculated using Equation (5):(5)$$\frac{1}{f\left(z\right)}=\frac{aLE_{p}f_{L}{}^{3/2}}{\omega \left(z\right)}\left(1-\frac{1}{\sqrt{N_{p}}}\right)\;$$
where $E_{p}$ is the energy per pulse (which ranged from 10 to 20 nJ, based on the average power), $f_{L}$ is the repetition rate (80 MHz), L is the sample thickness, $N_{p}$ is the number of pulses per sample and, $a=\alpha \left(dn/dT\right)/2\kappa {\left(\pi^{3}\delta \right)}^{1/2}$ is the fitting parameter. Additionally, dn/dT is the temperature derivative of the refractive index, κ is the thermal conductivity, and δ is the thermal diffusion coefficient.

The beam radius ω(z) of the samples was related to the laser beam waist $\omega_{o}$ and was calculated using Equation (6):(6)$$\omega \left(z\right)=\omega_{o}\sqrt{1+{\left(\frac{z}{z_{o}}\right)}^{2}}\;$$
where $z_{o}=\pi n\omega_{o}{}^{2}/\lambda$ is the Raleigh length of the beam.

Equation (7) was used to calculate the normalized transmittance of $T_{CA}$ [43,44,45,46,47,48], which was based on the thermal focal length f(z), where *z* is the relative position of the sample:(7)$$T_{CA}=1+\frac{2z}{f\left(z\right)}\;$$

Using Equation (8), the nonlinear refractive index $n_{2\;}$ of the Au NP samples could be estimated by knowing the on-axis nonlinear phase shift $\Delta \phi =\frac{z_{o}}{2f\left(0\right)}\;$, where f(0) is the focal length of the induced thermal lens when the sample is at z=0:(8)$$n=\frac{\lambda \omega_{o}{}^{2}\Delta \phi }{2P_{peak}L_{eff}}\;$$
where $P_{peak}$ is the peak power and $L_{eff}$ is the effective length of the sample. The change in $n_{2}$ with respect to the laser average power is shown in Figure 13, while the change in $n_{2}$ with respect to the excitation wavelength is shown in Figure 14.

It is worth mentioning that different reports have studied the nonlinear optical properties of Au NPs [19,49,50,51], but this work involved an intensive study that used different excitation wavelengths and different laser powers for two Au NP samples of different concentrations. This study allowed for a more complete understanding of the behavior of Au NPs for future applications in optical and photonic devices.

Based on the previous studies on the nonlinear optical properties of Au NPs [18,19,22,23,24,52,53,54,55,56] and the current study, we could conclude that the NL behavior and third-order nonlinear parameters of our Au NP samples were strongly dependent on the nanoparticle production method, as well as the laser parameters (such as wavelength, reputation rate, pulse duration, and peak intensity).

## 4. Conclusions

Two different Au NP samples were prepared using the laser ablation technique from a bulk high-purity Au sample. The two samples, each with different concentrations of Au NPs, showed quite narrow distributions of nanoparticles. The nonlinear absorption coefficient and nonlinear refractive index values of the samples were investigated using OA z-scan and CA z-scan approaches, respectively, in which the samples were irradiated using a femtosecond pulsed laser. The results showed that while the excitation wavelength was quite far from the SPR wavelength of the Au NPs, the samples exhibited strong RSA behavior that increased as the wavelength became longer. Moreover, the material showed a self-defocusing behavior when the CA z-scan technique was applied. These findings implied that Au NPs could be used to develop optical limiter materials with responses that are wavelength- and power-dependent.

## Figures and Tables

**Figure 1 nanomaterials-12-02980-f001:**
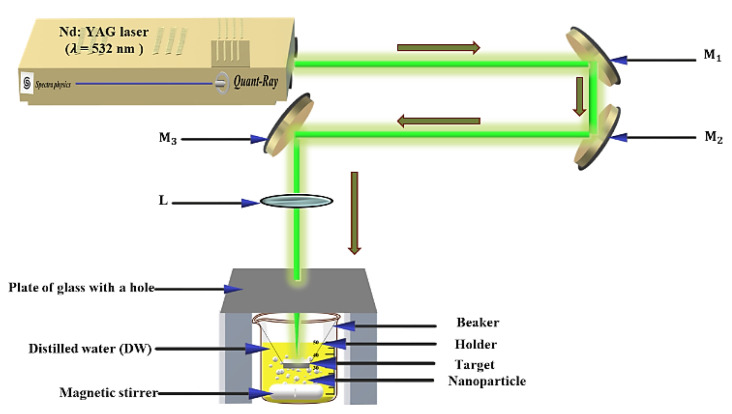
The experimental setup of the laser ablation method for the synthesis of the Au NPs using a nanosecond Nd: YAG Laser.

**Figure 2 nanomaterials-12-02980-f002:**
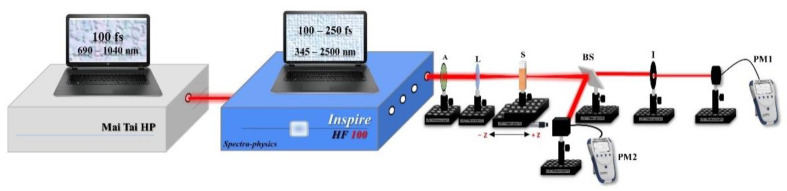
The experimental setup of the z-scans: A, attenuator; L, convex lens; S, Au NP sample; BS, beam splitter; I, iris; PM, power meter.

**Figure 3 nanomaterials-12-02980-f003:**
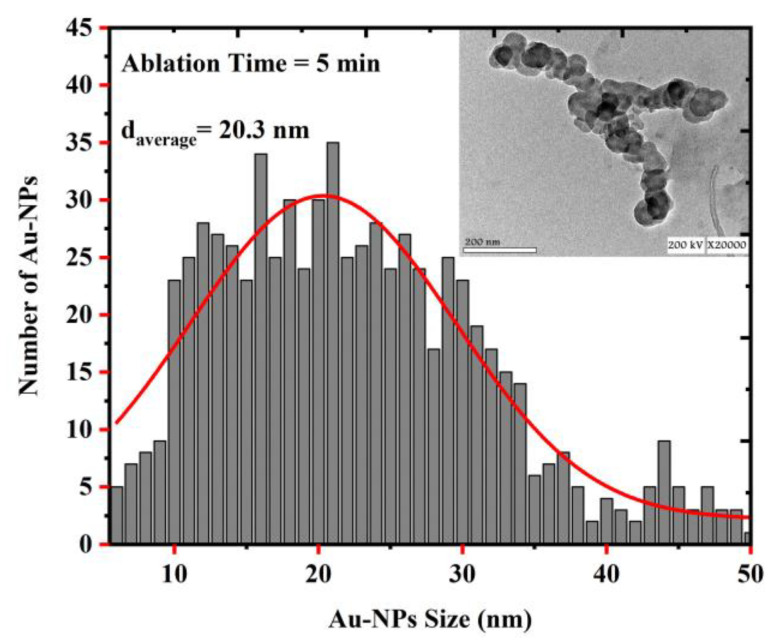
The size distribution of Au NPs that were prepared using a 5 min laser ablation time.

**Figure 4 nanomaterials-12-02980-f004:**
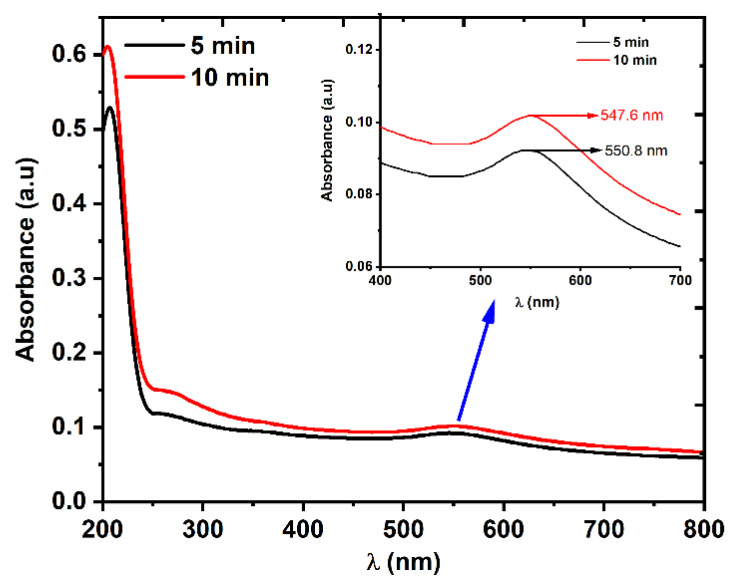
The size distribution of Au NPs that were prepared using a 10 min laser ablation time.

**Figure 5 nanomaterials-12-02980-f005:**
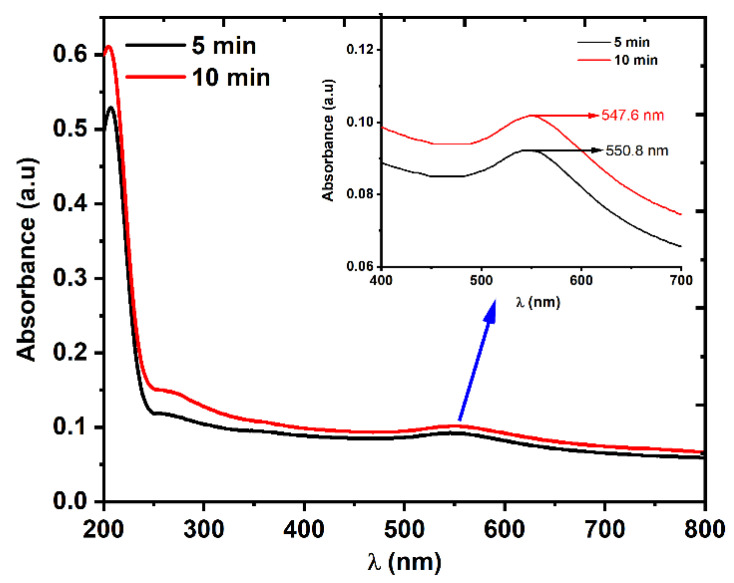
The linear absorption spectra of two Au NP samples that were prepared using 5 and 10 min laser ablation times. The LSPR peaks in the visible region are shown in the inset.

**Figure 6 nanomaterials-12-02980-f006:**
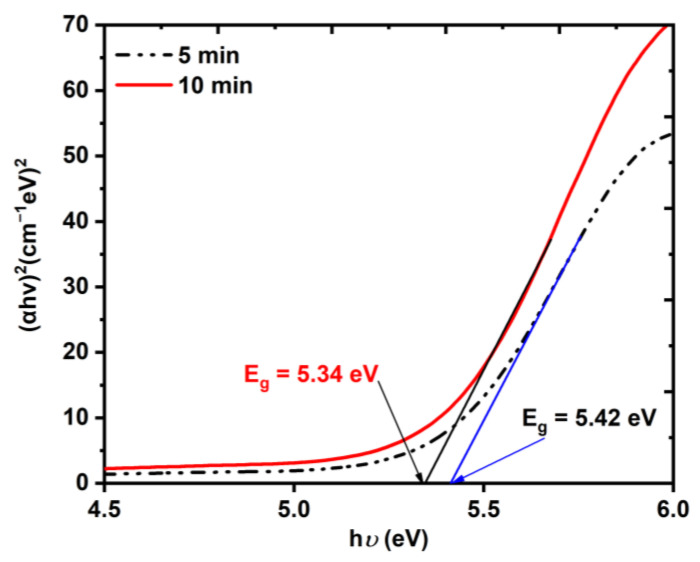
The energy bandgap of the Au NP thin film using the Tauc equation.

**Figure 7 nanomaterials-12-02980-f007:**
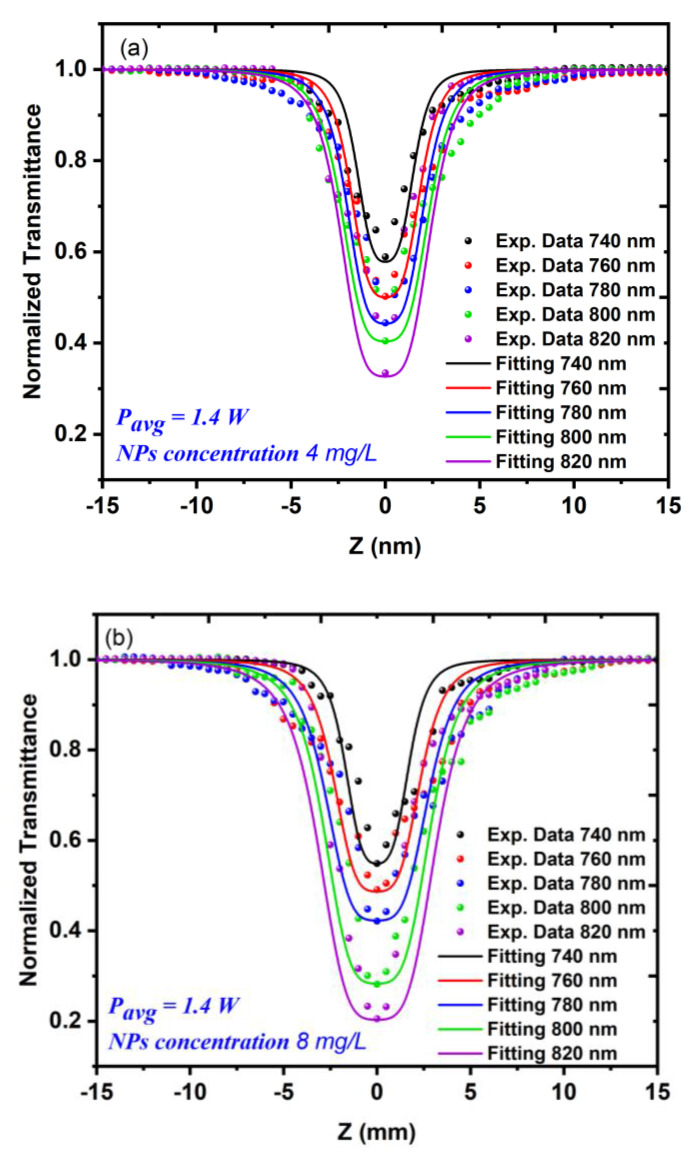
The OA z-scan measurements for the Au NP samples using different excitation wavelengths, which ranged from 740 to 820 nm at a constant average power of 1.4 W: (**a**) a sample concentration of 4 mg/L; (**b**) a sample concentration of 8 mg/L.

**Figure 8 nanomaterials-12-02980-f008:**
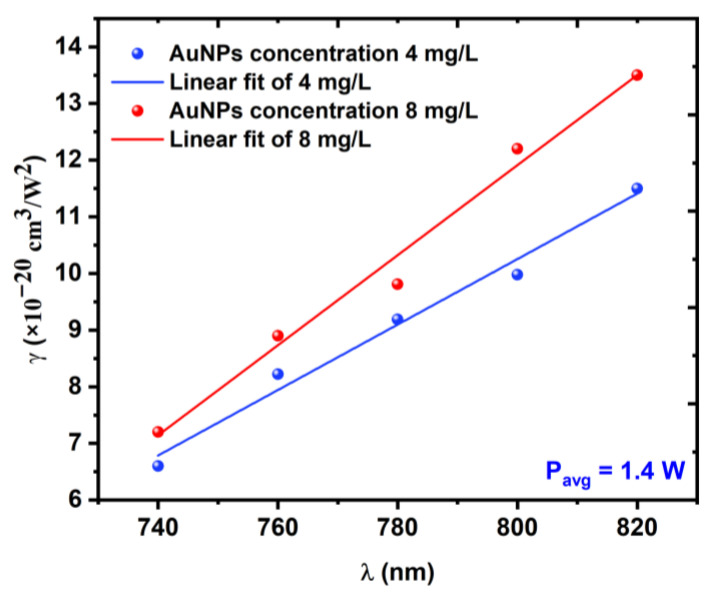
Variations in the NL absorption coefficient *γ* as a function of the excitation wavelength λ at a constant average power of 1.4 W.

**Figure 9 nanomaterials-12-02980-f009:**
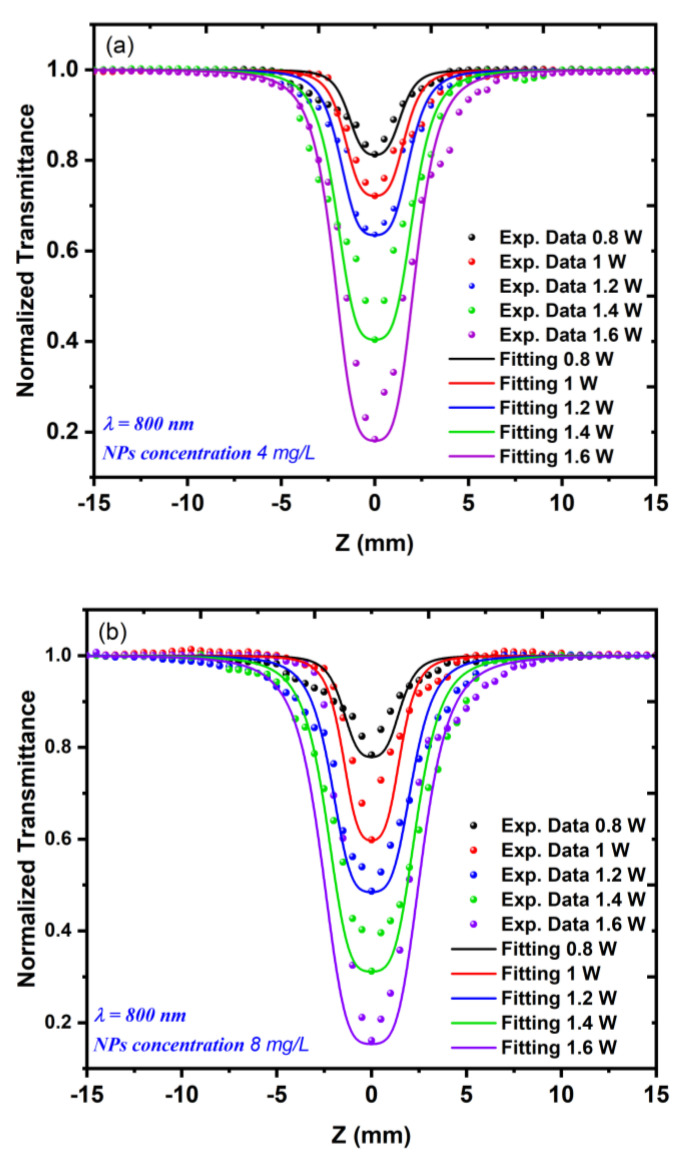
The OA z-scan measurements for the Au NP samples using different incident powers, which ranged from 0.8 to 1.6 W at a constant excitation wavelength of 800 nm: (**a**) a sample concentration of 4 mg/L; (**b**) a sample concentration of 8 mg/L.

**Figure 10 nanomaterials-12-02980-f010:**
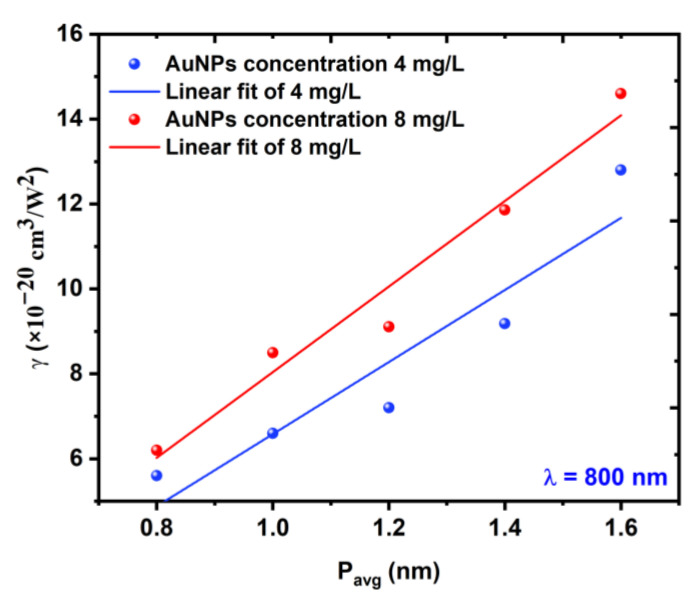
Variations in the NL absorption coefficient *γ* as a function of the average power $P_{avg}$ at a constant excitation wavelength of 800 nm.

**Figure 11 nanomaterials-12-02980-f011:**
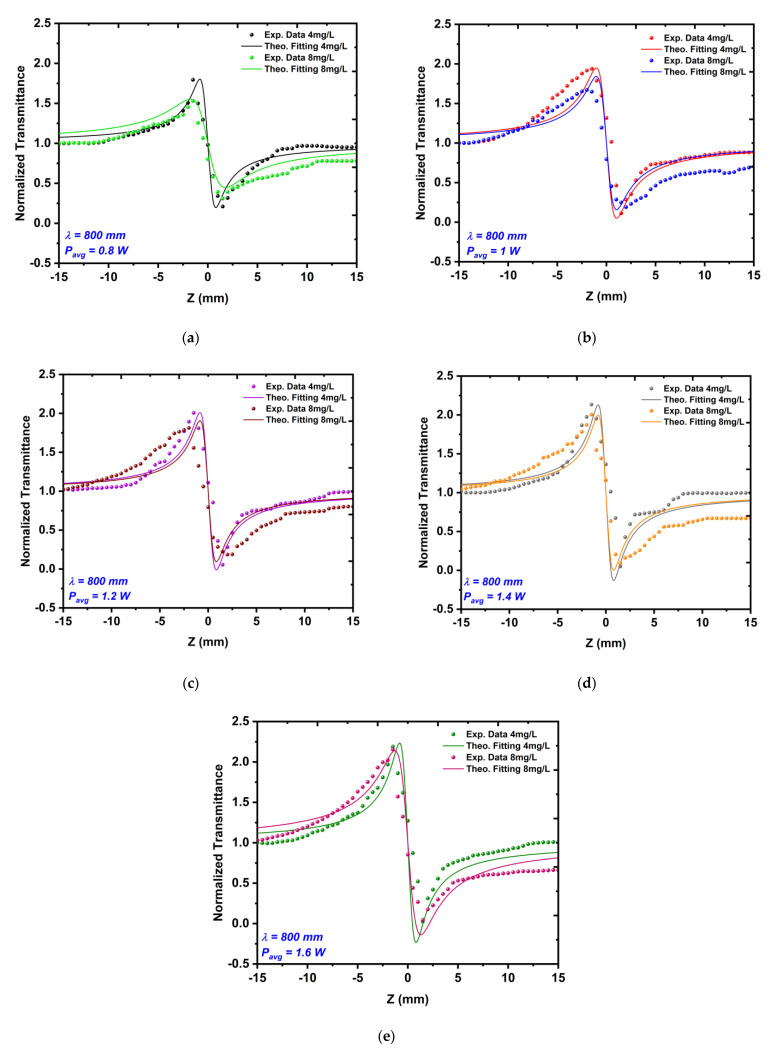
The CA z-scan measurements of the Au NP sample with a concentration of 4 mg/L for different average powers that ranged from 0.8 to 1.6 W (**a**–**e**) at a constant excitation wavelength of 800 nm.

**Figure 12 nanomaterials-12-02980-f012:**
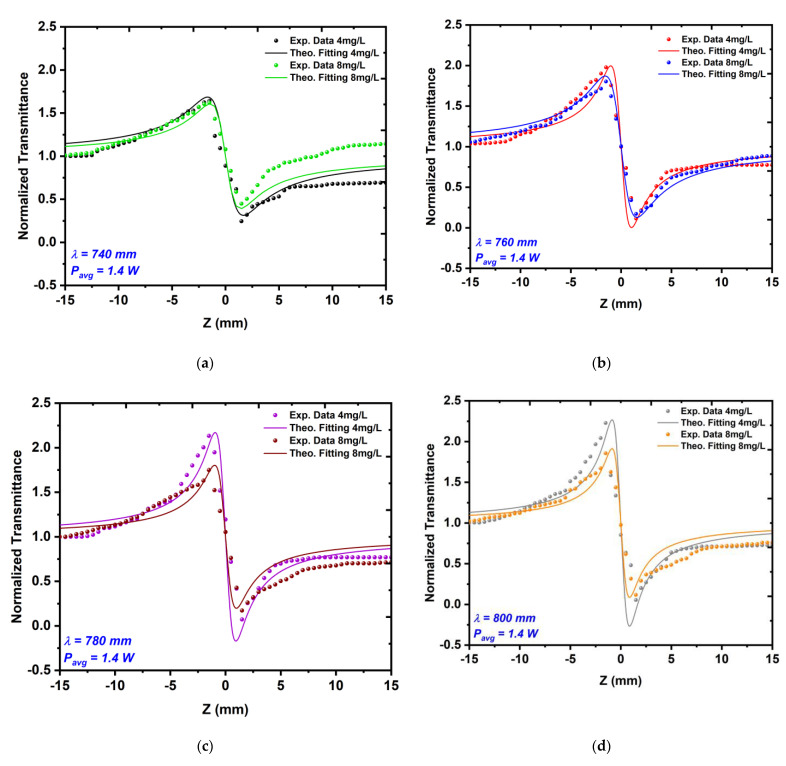
The CA z-scan measurements of the Au NP sample with a concentration of 8 mg/L for different excitation wavelengths that ranged from 740 to 820 nm (**a**–**e**) at a constant average power of 1.4 W nm.

**Figure 13 nanomaterials-12-02980-f013:**
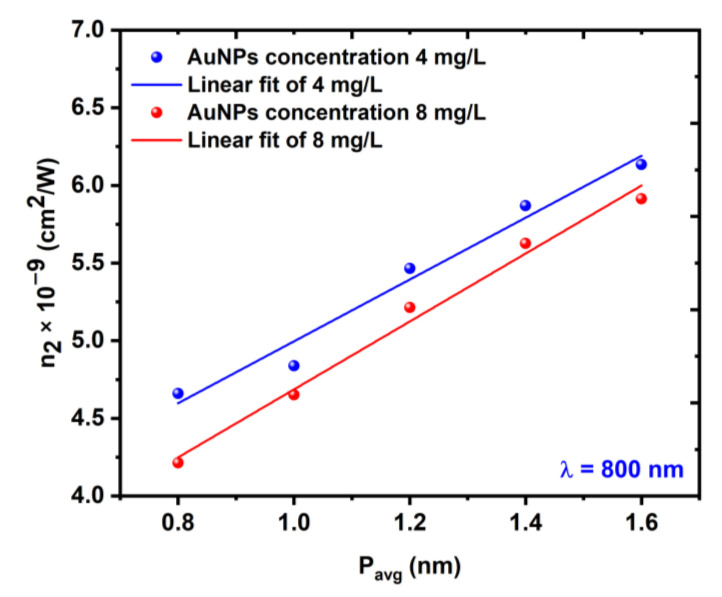
Variations in the NL refractive index *n*_2_ as a function of the average power $P_{avg}$ at a constant wavelength of 800 nm.

**Figure 14 nanomaterials-12-02980-f014:**
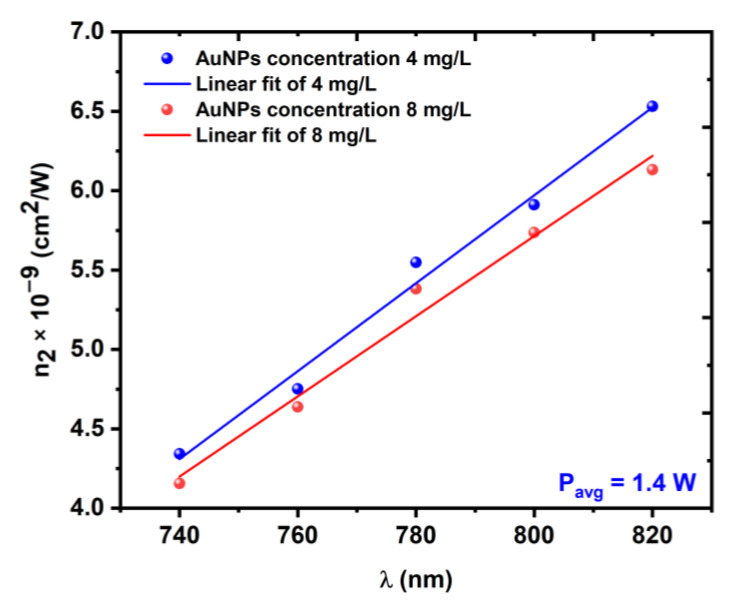
Variations in the NL refractive index *n*_2_ as a function of the excitation wavelength λ at a constant average power of 1.4 W.

**Table 1 nanomaterials-12-02980-t001:** The three-photon absorption cross-section for 4 mg/L and 8 mg/L Au NP concentrations.

$P_{avg}\;(W)$ at λ=800 nm	$\sigma_{3}\times 10^{-77}\;\;\left({\mathrm{cm}}^{6}{\;s}^{2}/{\mathrm{photons}}^{2}\right)$		λ (nm) $\mathrm{at}\;P_{avg}=1.4\;W$	$\sigma_{3}\times 10^{-77}\;\;\left({\mathrm{cm}}^{6}{\;s}^{2}/{\mathrm{photons}}^{2}\right)$	
	4 mg/L	8 mg/L		4 mg/L	8 mg/L
0.8	5.305	3.115	740	7.364	3.986
1	5.688	4.122	760	8.251	4.256
1.2	6.020	4.249	780	8.737	4.523
1.4	8.878	5.131	800	8.878	5.446
1.6	12.638	6.472	820	9.541	5.661

## Data Availability

The data that underlie the results that are presented in this paper are not publicly available at this time but can be obtained from the authors upon reasonable request.

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
