# Peer review of "Using Femtosecond Laser Pulses to Explore the Nonlinear Optical Properties of Au NP Colloids That Were Synthesized by Laser Ablation"

_nanomaterials, 2022, doi:10.3390/nano12172980_

Round 1

Reviewer 1 Report

The authors used the Z-Scan technique to investigate the nonlinear optical properties of gold nanoparticles fabricated by nanosecond laser ablation. It would be interested in areas of laser fabrication of nanomaterials and nanooptics. The paper can be accepted after major revision with considering the following comments.

1. the novelty of the paper should be clarified in the introduction part. Because, fabrication of nanoparticles by laser ablation have been widely investigated previously, including nanosecond laser and femtosecond laser. why the authors used the nanosecond laser ablation? what’s the difference between this paper and the previous articles?

2. Different laser parameters have important effects on the formation of nanoparticles, however, the authors only discussed the nanoparticle production at two different laser irradiation times in the paper. Fundamentally, more irradiation times need to be discussed to give a clear relationship between irradiation time and nanoparticle size and concentration. In addition, what about the influence of nanoparticle size and concentration with laser energy?

3. Figure 3 and 4, the TEMs of nanoparticles were unclear, especially, the scale bar in the TEM was too small to read.

4. Page 3, part 3.1, line 4, the “in with” should be rechecked.

5. Page 6, line 13, the ”5mg/L” should be changed to 8mg/L.

6. Figure 5, the peak near 550 nm was appeared due to surface plasmon resonance absorption? The peak position here was not obvious, and the authors need to redrawing the absorption spectrum at the peak position for a more obvious expression.

Author Response

Response to Reviewer 1 Comments

The authors used the Z-Scan technique to investigate the nonlinear optical properties of gold nanoparticles fabricated by nanosecond laser ablation. It would be interested in areas of laser fabrication of nanomaterials and nanooptics. The paper can be accepted after major revision with considering the following comments.

Point 1: The novelty of the paper should be clarified in the introduction part. Because, fabrication of nanoparticles by laser ablation have been widely investigated previously, including nanosecond laser and femtosecond laser. why the authors used the nanosecond laser ablation? what’s the difference between this paper and the previous articles?

Response 1: We appreciate the reviewer's input. The introduction section of the revised manuscript now includes a discussion of the novelty of the present study (highlighted in red)  as follows:

“Although there have been a few reports on the investigation of NLO properties of Au-NPs using the Z-scan approach [18-24], no systematic study of the NLO of Au-NPs colloids employing femtosecond laser pulses at different Au-NPs sizes and concentrations, and different laser parameters such as excitation wavelengths and excitation intensities has been undertaken to our knowledge. For instance, the [18-20, 22, 23]  studied the NLO  properties of Au-NPs at a certain excitation wavelength and at a given Au-NPs size. In [24], the NLO properties of Au-NPs at various concentrations and a certain excitation wavelength were studied”.

References

  1. Ganeev, R. A., Suzuki, M., Baba, M., Ichihara, M., & Kuroda, H. (2008). Low-and high-order nonlinear optical properties of Au, Pt, Pd, and Ru nanoparticles. Journal of Applied Physics, 103(6), 063102.
  2. Souza, R. F., Alencar, M. A., da Silva, E. C., Meneghetti, M. R., & Hickmann, J. M. (2008). Nonlinear optical properties of Au nanoparticles colloidal system: local and nonlocal responses. Applied Physics Letters, 92(20), 201902.
  3. Bigot, L., El Hamzaoui, H., Le Rouge, A., Bouwmans, G., Chassagneux, F., Capoen, B., & Bouazaoui, M. (2011). Linear and nonlinear optical properties of gold nanoparticle-doped photonic crystal fiber. Optics express, 19(20), 19061-19066.
  4. Trejo-Durán, M., Cornejo-Monroy, D., Alvarado-Méndez, E., Olivares-Vargas, A., & Castano, V. M. (2014). Nonlinear optical properties of Au-nanoparticles conjugated with lipoic acid in water. Journal of the European Optical Society-Rapid publications, 9.
  5. Tajdidzadeh, M., Zakaria, A. B., Talib, Z. A., Gene, A. S., & Shirzadi, S. (2017). Optical nonlinear properties of gold nanoparticles synthesized by laser ablation in polymer solution. Journal of Nanomaterials, 2017.
  6. Krishnakanth, K. N., Bharathi, M. S. S., Hamad, S., & Rao, S. V. (2018, April). Femtosecond nonlinear optical properties of laser ablated gold nanoparticles in water. In AIP Conference Proceedings (Vol. 1942, No. 1, p. 050122). AIP Publishing LLC.
  7. AL-Hamdani, A. H., A. Madlool, R., & Abdulazeez, N. Z. (2020, December). Effect of gold nanoparticle size on the linear and nonlinear optical properties. In AIP Conference Proceedings (Vol. 2290, No. 1, p. 050029). AIP Publishing LLC.

Point 2: Different laser parameters have important effects on the formation of nanoparticles, however, the authors only discussed the nanoparticle production at two different laser irradiation times in the paper. Fundamentally, more irradiation times need to be discussed to give a clear relationship between irradiation time and nanoparticle size and concentration. In addition, what about the influence of nanoparticle size and concentration with laser energy?

Response 2: We appreciate the feedback from the reviewers. We will follow the Reviewer's recommendation in our future works with utmost care. Moreover, in this study, we focused more on the effect of laser parameters on the NLO properties of the AU-NPs.

Point 3: Figure 3 and 4, the TEMs of nanoparticles were unclear, especially, the scale bar in the TEM was too small to read.

Response 3: We appreciate the reviewer remarks, which have already been incorporated into the revised manuscript. Figures 3 and 4 have been replaced with new figures that depict the scale bar in the TEM images.

Point 4: Page 3, part 3.1, line 4, the “in with” should be rechecked.

Response 4: We appreciate the reviewer's input. The typos have been checked and corrected in the updated manuscript (marked in red).

Point 5:  Page 6, line 13, the ”5mg/L” should be changed to “8mg/L”.

Response 5: We appreciate the reviewer's input. The typos have been checked and corrected in the updated manuscript (marked in red).

Point 6:  Figure 5, the peak near 550 nm was appeared due to surface plasmon resonance absorption? The peak position here was not obvious, and the authors need to redrawing the absorption spectrum at the peak position for a more obvious expression.

Response 6: We appreciate the reviewer remarks, which have already been incorporated into the revised manuscript. Figure 5 has been replaced with a new figure that depicts the surface plasmon resonance absorption peaks at 550 nm as an inset in Fig. 5.

Reviewer 2 Report

In this paper, the authors study the nonlinear optical properties of AuNPs after laser ablation. They first demonstrate how they fabricate the AuNPs through laser ablation. Then they conduct a series of studies (i.e. wavelength, laser power, ablation time, etc.) on the linear and nonlinear properties of fabricated AuNPs. I think this work is complete and well written. However, there are still some minor issues in this paper. Therefore, this manuscript needs minor revision. Below are my comments.

1.     Can the author provide more details about how they fabricate the AuNPs? Is it fabricated using AuCl3 solutions?

2.     How does the heating generated by the fs laser during the measurement affects the optical properties of AuNPs? Will the heating generated by the fs laser change the sizes of the nanoparticles, which makes the TEM measurements (Figs. 3,4) and linear absorption (Fig.5) invalid?

Minor issues:

The author mentioned surface plasmon resonance (SPR). It typically refers to the surface plasmon effect of Au film or metasurface. I prefer to use localized surface plasmon resonance (LSPR) as the right term for AuNPs. 

Author Response

Response to Reviewer 2 Comments

In this paper, the authors study the nonlinear optical properties of AuNPs after laser ablation. They first demonstrate how they fabricate the AuNPs through laser ablation. Then they conduct a series of studies (i.e. wavelength, laser power, ablation time, etc.) on the linear and nonlinear properties of fabricated AuNPs. I think this work is complete and well written. However, there are still some minor issues in this paper. Therefore, this manuscript needs minor revision. Below are my comments.

Point 1: Can the author provide more details about how they fabricate the AuNPs? Is it fabricated using AuCl3 solutions?

Response 1: We appreciate the reviewer's input.  The Au-NPs in this study were created utilizing a laser ablation process described in section 2.1, in which a Nd: YAG laser was employed to generate the Au-NPs from 99% pure bulk Au submerged in a glass beaker filled with 22 ml of distilled water.

Point 2: How does the heating generate by the fs laser during the measurement affects the optical properties of AuNPs? Will the heating generated by the fs laser change the sizes of the nanoparticles, which makes the TEM measurements (Figs. 3,4) and linear absorption (Fig.5) invalid?

Response 2: We appreciate the reviewer’s remark. The TEM measurements and linear absorption depicted in Figs 3, 4, and 5 were performed immediately after the Au-NPs were created using a Nd: YAG laser with a repetition rate of 10 Hz, so there was no heat influence on such measurements. Furthermore, due to the use of a high repetition rate fs laser ( 80 MHz) during the measurement of the NLO properties of the Au-NPs, the heat generated during the experiment, which may lead to a thermal lens, has been well treated using the mathematical model that is explained in section 3.3.2, where we take into account the number of pulses and the experiment time during calculations.

Point 3: Minor issues:The author mentioned surface plasmon resonance (SPR). It typically refers to the surface plasmon effect of Au film or metasurface. I prefer to use localized surface plasmon resonance (LSPR) as the right term for AuNPs. 

Response 3: We appreciate the reviewer's feedback. The SPR has been replaced with the LSPR in the revised manuscript (marked in red).

Round 2

Reviewer 1 Report

In figure 3 and 4, the text in the TEM image was still too small to read. 

Author Response

Response to Reviewer 1 Comments

The authors used the Z-Scan technique to investigate the nonlinear optical properties of gold nanoparticles fabricated by nanosecond laser ablation. It would be interested in areas of laser fabrication of nanomaterials and nanooptics. The paper can be accepted after major revision with considering the following comments.

Point 1: In figure 3 and 4, the text in the TEM image was still too small to read.

Response 1: We appreciate the reviewer remarks, which have already been incorporated into the revised manuscript. Figures 3 and 4 have been replaced with new figures that depict the scale bar in the TEM images.
